# Opinion: Are Organoids the End of Model Evolution for Studying Host Intestinal Epithelium/Microbe Interactions?

**DOI:** 10.3390/microorganisms7100406

**Published:** 2019-09-29

**Authors:** Michelle M. George, Mushfiqur Rahman, Jessica Connors, Andrew W. Stadnyk

**Affiliations:** 1Department of Pediatrics, Division of Gastroenterology & Nutrition, Dalhousie University, 5850 University Avenue, Halifax, NS B3K 6R8, Canada; mc277205@dal.ca (M.M.G.); mushfiqur.rahman@dal.ca (M.R.); jessica.connors@iwk.nshealth.ca (J.C.); 2Department of Microbiology & Immunology, Dalhousie University, 5850 College Street, Halifax, NS B3H 4R2, Canada; 3Department of Pathology, Dalhousie University, 5850 College Street, Halifax, NS B3H 4R2, Canada

**Keywords:** organoid, enteroid, colonoid, intestinal epithelium, stem cell, Lgr5, Paneth cell, norovirus, rotavirus, *Salmonella*

## Abstract

In the pursuit to understand intestinal epithelial cell biology in health and disease, researchers have established various model systems, from whole animals (typically rodents) with experimentally induced disease to transformed human carcinomas. The obvious limitation to the ex vivo or in vitro cell systems was enriching, maintaining, and expanding differentiated intestinal epithelial cell types. The popular concession was human and rodent transformed cells of mainly undifferentiated cells, with a few select lines differentiating along the path to becoming goblet cells. Paneth cells, in particular, remained unculturable. The breakthrough came in the last decade with the report of conditions to grow mouse intestinal organoids. Organoids are 3-dimensional ex vivo “mini-organs” of the organ from which the stem cells were derived. Intestinal organoids contain fully differentiated epithelial cells in the same spatial organization as in the native epithelium. The cells are suitably polarized and produce and secrete mucus onto the apical surface. This review introduces intestinal organoids and provide some thoughts on strengths and weaknesses in the application of organoids to further advance our understanding of the intestinal epithelial–microbe relationship.

## 1. The Advent of Organoids in the Study of Epithelial Cells and Microbes

The average human has as many as 3 × 10^13^ bacteria in/on their body, of which trillions are constituents of the intestinal microbiome [1]. These bacteria and other organisms in the microbiome interface with epithelial cell barriers of the host. Thus, the microbial impact on host physiology and metabolic processes in achieving homeostasis are governed to a large extent by epithelial cells [2]. Over the history of investigation to understand this microbial/epithelial cell relationship, a variety of intestinal epithelial cell model systems have been developed, even evolved, as investigators worked to close the gap between perceived deficiencies in the models and our understanding of the native epithelium. Whole animal models included ligated loops of rodent intestines, direct treatment of rodents with induced experimental intestinal injury, and manipulations like human fetal xenotransplanted intestines in immune-deficient mice (Figure 1). Intestinal cell line studies included (but were not limited to) human fetal and germ-free rodent primary cell cultures and transformed carcinoma lines, which have been very popular and have led to rich insights into the host–microbe relationship. However, these transformed cell lines have many limitations, especially from a translational perspective. One limitation is the lack of mature epithelial cell types that can be differentiated from cell lines. Another limitation is that most of the available intestinal epithelial cell lines proliferate when subconfluent then stop dividing when confluent, whereas cells in the native epithelium proliferate while “confluent”, possessing all (or more) of the cell–cell adhesive interactions of the confluent cell lines. The hope for more phenotype features in ex vivo epithelial cell cultures has thrust forward with the discovery a decade ago of conditions to grow 3-dimensional cell cultures from adult intestines that have similar cellular spatial organization and dividing and differentiated cell types as the host species epithelium from which the cultures were derived [3,4]. In this regard, intestinal organoids—enteroids (small intestine) and colonoids (large intestine/colon)—provide a more representative tool for the study of intestinal epithelial development, physiology, and diseases [5,6]. Enteroids and colonoids are comprised of epithelial cells that have differentiated and self-organized into mature sub-types including enterocytes, Paneth cells, goblet cells, enteroendocrine cells, and even Tuft cells [3]. Prior to this achievement, most fully differentiated cell types from adult intestines could not be grown or maintained in culture, nor were most cell systems roofed by the mucus typical of the intestines. In the intestines, epithelial cells are essentially exposed to microbial filtrates with the majority of microbes kept “at bay” by a mucus layer. Intestinal organoids suitably reproduce these conditions. Having recapitulated the in vivo conditions, ex vivo organoids hold great potential to refine our understanding of the epithelial response to microbes. There are multiple recent reviews on organoids, which consider specifics of the engineering requirements for model epithelial culture conditions [7], review intestinal organoids and infectious microbial interactions [8], or focus on *Salmonella* infection [9]. Here, we offer thoughts on the strengths and weaknesses of intestinal organoids in addressing the following question: Are organoids the ultimate stage of intestinal cell model evolution?

## 2. Cell Biology of Organoids

Not limited to the intestines, organoids are generally defined as three-dimensional assemblages of cells that grow from organ-specific progenitors or stem cells and resemble a miniaturized ex vivo organ from which they have been derived [10,11]. Currently, the primary approaches to generating intestinal organoids include isolating fresh crypts that contain multipotent adult stem cells (ASC) or using embryonic or inducible pluripotent stem cells (iPSCs). In order to proliferate and differentiate into functioning organoids, stem cells require conditions and signals similar to in vivo niche exposures [12]. In the presence of these signals, stem cells differentiate into multiple cell types within a particular lineage, spatially self-organize to resemble the source organ, and importantly, are capable of conducting functions of the source organ including secretion, filtration, absorption, endocrine secretion, neural activity, and contraction. Some intestinal-specific functions will be described below. The ability to cultivate self-renewing organoids has been propelled by recent breakthroughs in the identification and development of critical synthetic growth factor cocktails specific to different tissue niches [13].

By including multiple cell types of the organ from which the stem cells were derived, organoids have a unique potential to model human disease. They are self-renewing and can be readily cryopreserved (which obviates the need for continued access to primary tissue), and they can be genetically and environmentally manipulated in highly controlled settings [14]. Importantly, the prospect of expanding and maintaining organoids from individual patients introduces the potential for personalized medicine, testing tissue-directed treatments against particular genotypes for achieving optimal outcomes for single patients.

## 3. Native Intestinal Epithelial Cell Organization 

The human intestine is lined with a single-cell layer of columnar epithelia that forms an essential barrier to environmental challenges arising in the lumen. To maintain this barrier, especially in the context of pathogen- or stress-induced tissue damage, intestinal epithelial cells possess a high rate of self-renewal. This renewal process begins with intestinal stem cells (ISCs) residing in intestinal crypts that give rise to rapidly dividing transit-amplifying (TA) cells which form the rest of the crypt and flow up the villus where they differentiate into various epithelial cell subtypes and eventually die. There are two populations of ISC in the adult intestinal crypt: a quiescent ISC (also called “+4 cells” due to their position in the crypt) and actively proliferative ISCs, which are characterized by expression of Lgr5 [15,16]. To generate organoids from adult ISCs, native epithelium is dissociated into single crypts from which Lgr5^+^ ISCs organize into crypt-like structures after receiving niche signals, most notably Wnt3 and Epidermal Growth Factor (EGF), from Paneth cells sandwiched between ISCs in the crypt [17,18]. For crypts to grow and proliferate, niche factors must be added to the culture. In the artificial context of a plastic dish, the basal lamina is replaced by a laminin-rich extracellular matrix plug (e.g., Matrigel^®^) and a cocktail of the niche factors R-spondin, EGF, and Noggin is provided in culture media. Wnt is a critical factor for maintaining stemness and for driving the proliferation of TA cells. Wnt normally occurs in a spatial gradient, being most concentrated in the crypt base. In the absence of Wnt, near the top of the crypt, TA cells differentiate into epithelial subtypes such as enterocytes, goblet cells, enteroendocrine cells, tuft cells, and M cells [19]. Depending on the source of the crypt, additional factors may be necessary for growth. For instance, Paneth cells are believed to be a sufficient source of Wnt in small intestinal crypts, whereas colon crypts, which lack Paneth cells, must have Wnt supplemented in growth media [19]. Notably, organoids from both stem cell sources are amenable to transfection and CRISPR/Cas9 gene editing.

## 4. Intestinal Organoids Closely Resemble Native Epithelium 

Both organoids and the human gut epithelium are comprised of multiple cell types; ISCs differentiate into these, including the abundant enterocytes, goblet cells, Paneth cells, enteroendocrine cells and Tuft cells. Differentiation of cells within the enteroids is hallmarked by products commonly associated with cells in vivo such as alkaline phosphatase and sucrase-isomaltase in enterocytes, mucins in goblet cells, and lysozyme in Paneth cells [5,20]. Human intestinal epithelial cells also exhibit different properties, often expressed in a cell polarized fashion, along the cephalad-caudal length of the major segments including duodenum, jejunum, ileum, and colon. For example, the SGLT1/SLC5A1 transporter is more abundant in the duodenum than the ileum [21]. Importantly, cells within organoids retain many of these markers including ion transport systems, for example, the apically expressed sodium/glucose transporter 1 (SGLT1/SLC5A1), proton-coupled peptide transporter (PEPT1/SLC15A1), and basally expressed facilitated glucose transporter (GLUT2) [22]. Another example of cytoplasmic molecules in epithelial cells that show gradients of expression along the lengths of the intestines are the aquaporins [23], and examining aquaporins may provide more insights into whether cell phenotypes are absolutely retained depending on the region from which the stem cells were derived. So, while some of these properties might be presumed to be shaped by exposures to exogenous stimuli absent in organoid cultures, there is evidence that phenotypes are intrinsic to the region of intestine from which the organoids were derived [24].

## 5. Differences Between Native Epithelium and Organoids

Notwithstanding the evidence that cell phenotypes are programed into the local stem cell, enteroids and colonoids lack epithelial-adjacent tissues including enteric nerves, vasculature, gut-associated lymphoid tissues, luminal food products and interface with the microbiome, all of which may provide extrinsic signals to epithelial cells. In addition to food, presumably, the native intestinal epithelium is bathed in a filtrate of microbial products, which is not commonly emulated in standard organoid culture conditions. Yet it is unequivocal that microbial products impact epithelial cell physiology. One example of a microbial product that when added to organoid cultures clearly modified development is the TLR3 agonist poly I:C. Exogenous poly I:C increased enteroid surface area but reduced colonoid crypt budding associated with reduced numbers of lysozyme positive cells in both types of intestinal organoid [25]. A differential impact of poly I:C on enteroids versus colonoids was also registered at the transcript level, including some inflammatory mediators and stem cell markers [25]. This differential effect on enteroids versus colonoids is compatible with the understanding that there are cephalad -caudal distinctions in intestinal epithelial cells, but does it also mean intestinal organoids should be grown in the presence of a microbial product cocktail? 

Cells becoming polarized is critical to studying microbial infection of intestinal epithelial cells. The application of microbes or microbial products limited to the apical cell surface is challenging in organoid cultures since while the cells in organoids polarize, the apical cell surface faces a closed “lumen” (resulting in secreted products and dead cells accumulating in the lumen). A recent noteworthy adaptation that addresses access to the apical membrane was reported by Co et al. (2019) [26]. Acting on the understanding that matrix proteins promote the polarity achieved in enteroids, the authors digested the matrix “bubble” and continued the culture as a suspension, to discover that the enteroid cells reversed polarity such that the apical membrane was no longer facing the closed lumen but was on the “outside” [26]. Remarkably, the early “apical-out” organoids continued to mature to include the polarized differentiated cell types. It remains to be determined whether the cells respond to microbial products in a manner resembling “apical-in” organoids, but this discovery certainly enhances the opportunity to study apical microbial challenges.

Organoids generated from fresh human tissue and therefore derived from ASC, possess cell types and important physiological properties of the organ from which they were obtained. On the other hand, there is great interest in the potential for using human embryonic and iPSCs to generate organoids in order to circumvent the scarce availability and accessibility to primary human tissue samples; however, there is compelling evidence that intestinal organoids derived from iPSCs are similar to fetal human intestine in terms of expressed progenitors and genes [27]. Finkbeiner and co-workers compared the transcriptional profile of human pluripotent stem cell-derived enteroids and fetal small intestine with adult small intestine and found that genes involved in development are enriched in both fetal intestine and iPSC-derived enteroids, whereas genes involved in digestive function and Paneth cell host defense are expressed at higher levels in adult tissue [28]. Notably, iPSC-derived enteroids undergo further maturation to more closely resemble the adult intestine following in vivo transplantation. Thus, important maturation events occur in the intestine. Whether factors mentioned above as lacking in organoid cultures would result in cell transcriptomes more closely resembling adult cells remains to be tested.

## 6. Intestinal Organoids Overcome Earlier Model Limitations 

Human diseases are often modeled in rodents. According to the U.S. Secretary of Health and Human Services, “Nine out of ten experimental drugs fail in clinical studies because we cannot accurately predict how they will behave in people based on laboratory and animal studies” (FDA (2006) FDA Issues Advice to Make Earliest Stages of Clinical Drug Development More Efficient. http://www.fda.gov/NewsEvents/Newsroom/PressAnnouncements/2006/ucm108576.htm). The availability of organoids provides a model system to study and test clinically relevant drugs. Certainly, using organoids for safety screening is likely to be more informative than previous approaches relying on 2-D monolayer cell lines that do not have the complex cell types and interactions present in vivo. Primary cell lines and tissue biopsies, although possibly preserving relevant cell signaling pathways and physiological properties, like cell lines are at risk of contamination by other cells, change ploidy, and mutate and/or change phenotype after being passaged over an extended period of time [29]. Organoids retain the genotype of the donor cells and thus allow for modelling the intestinal physiology of genetically linked diseases such as inflammatory bowel disease, cystic fibrosis, and colorectal cancer. Organoids derived from individual donors allow researchers to explore phenotypes for specific patients tailored to their unique genotype. This should prove especially important for the development of gene-targeted therapies.

One such personalized, donor-specific application is using organoids as a cell source for tissue regeneration. Demonstrated in mice, colonoids have been grown from Lgr5^+^ stem cells and engrafted in damaged colonic tissue in vivo, with engrafted mice ultimately achieving higher body weights than the controls [30]. This approach overcomes the histocompatibility disparity that would otherwise require immunosuppression should allogeneic donor cells are used for tissue repair.

Organoids also enable the study of the interaction between anaerobic bacteria and the human gut. Cell lines are grown in standard O_2_ levels and do not provide a suitable environment for studying anaerobic bacteria living in the gut. Organoids are grown in an extracellular matrix plug and can be used to establish an O_2_ gradient that will allow the bacteria to survive [31]. Yet despite this control over O_2_ availability, another refinement of monolayer cell culture approaches related to oxygen availability is to establish an air–liquid interface on the apical aspect of the cells. Using a porcine cell line, it was reported that growing cells with an air–liquid interface had a profound effect on cell physiology, principally leading to enhanced oxidative phosphorylation and suppressed glycolysis [32]. This has not been achieved using organoids, but dissolution of the organoid into monolayers may benefit from this adaptation.

## 7. Organoids and Microbes

Along the evolutionary path of intestinal epithelial cell models, investigators have eagerly applied their model to better understand host interactions with pathogenic or commensal microbes. Consequently, much of our understanding of the epithelial cell response is drawn from experiments using transformed and undifferentiated cells and these were challenged with incompatibilities with the infectivity of organism, reducing confidence that outcomes can be applied to the living human. Notwithstanding the shortcoming discussed previously, organoids offer new opportunities for answers to questions about mechanisms of infection and host response. First, organoids can affirm (or dispute) our understanding of intestinal epithelial biology established from transformed cells. Second, using organoids allows specific examination of fully differentiated non-dividing versus dividing non-differentiated epithelial cell response to microbial challenges. Lastly, organoids permit an in-depth examination of unique (patient or engineered) genotype–phenotype associations in responses to microbes, which is highly relevant to disorders thought to arise from abnormal host–microbe interactions, such as the inflammatory bowel diseases. However, a challenge remains in accessing the apical side of the cells where microbes would typically encounter epithelial cells, which some investigators have overcome using microinjections and the “apical outside”, described earlier. Here, we review a few examples of research that highlight how using organoids is re-writing our understanding of microbe/intestinal epithelial cell interactions.

### 7.1. Host–Virus Research Has Benefited from Enteroids: Rotavirus and Norovirus

Enteric viruses, like the human noro and rota viruses, are among the most significant causes of acute gastroenteritis; combined, they are responsible for approximately 70% of episodes [33]. What is known about the pathophysiology of human rotavirus (HRV) was derived mainly from in vitro studies conducted using simian rotavirus or simian agent 11 in transformed cells or in monkey kidney cell lines. HRVs generally replicate poorly in transformed cell lines as they have a restricted host range, which also prevents them from replicating in most non-primate animal models. Enteroids introduced a new platform for studying HRV pathogenesis in the preferred host. Enteroids from different patients are readily infected by and also support the full productive, rapid, and lytic replication process of HRV strains [34]. Infecting the enteroids with HRV also revealed that differentiated cells support a considerably higher viral growth than undifferentiated cells [34]. The model also showed that although HRV predominantly infected differentiated enterocytes, the virus also infected enteroendocrine cells of the human small intestine. Rotaviruses vary in the levels of replication in enteroids grown from different patients, indicating that there is a difference in susceptibility between individuals. Enteroids may also provide a platform for evaluating antivirals aimed at chronically infected transplant patients [35]. 

The innate response mounted by the intestinal epithelium against viral infections is critical in controlling and possibly lowering the viral load during the initial infection. When exposed to HRV infection, enteroids mounted an innate immune response that mainly activated transcriptional pathways leading to production of type III interferons (IFNs). However, the induced IFNs were insufficient to curtail the HRV infection while adding type I IFNs to the system was successful in restricting viral infection [36]. However, this was not the case in the enteroids and HRV replication was not repressed by the type III IFNs. This discrepancy between what is predicted and what is observed highlights the importance of using enteroid models to give a more accurate representation of the role innate immunity mounted by epithelial cells plays in infection management.

Prior to work conducted using organoids, our understanding was that human norovirus strains replicated in B cells, but since the virus can infect B cell-deficient patients, there must be another host cell type [37]. Consequently, enteroids have been used to demonstrate that norovirus infects intestinal epithelial cells. Much like HRV, earlier efforts at cultivating norovirus in transformed cells have been unsuccessful; however, these fastidious viral strains thrive in human intestinal organoid and monolayer enteroid cultures. Human norovirus replicates in duodenal, jejunal, and ileal enteroids, but replication varies by strain and intestinal segment. In fact, replication and growth has only been detected in enterocytes and not in goblet or enteroendocrine cells, suggesting that enterocytes are the primary cell target for infection. As noroviruses are genetically diverse, there are strain-specific requirements such as the addition of bile, needed to enhance replication in enteroids. This latter point emphasizes the reductionist model enteroids truly represent, despite the cellular complexity and physiology.

### 7.2. Host–Bacteria Research Has Benefited from Enteroids: Salmonella and E.coli

*Salmonella* spp. are gram-negative enteric pathogens that annually cause millions of cases of gastroenteritis and typhoid fever worldwide. *Salmonella* spp. invades intestinal epithelial cells via bacterial-mediated endocytosis by preferentially binding to the highly specialized M cells on Peyer’s Patches. Studies of the pathogenesis of *S. typhimurium* have commonly been undertaken in mice; however, S. *typhimurium* dissemination occurs in mice with the infection becoming systematic, while in humans the disease typically remains localized in the intestines [38]. Studies conducted in mouse enteroids have reinforced earlier observations that *Salmonella* disrupts tight junctions, that epithelial cells activate NF-κB as an inflammatory response to infection, and that there is a significant decrease in the number of stem cell marker Lgr5^+^ cells in enteroids [39]. Although studies using mouse enteroids support the reinforced previous observations, mouse cells lack the human genetic identities that are required to truly understand the infectivity pathway of the bacterium. Consequently, inducible human enteroids coupled with microinjection of the bacteria were shown to upregulate multiple inflammatory mediators and thus be a suitable model for studying host–bacterial pathogen interactions [40].

As an example of how pure epithelial cell cultures may not tell the entire story, including during infection, the intimate relationship between intestinal epithelial cells and intraepithelial lymphocytes (IELs) will be touched on here. In the healthy intestine, interleukin (IL)-7 made by epithelial cells acts on IELs [41], while IL-15 made by IELs impact epithelial cells [42], a model example of a juxtacrine cytokine network. Additionally, both cell types are responsive to microbial factors, with some IEL populations dependent on microbes in the lumen (reviewed in [43]). *Salmonella* infection of mice has exposed another reciprocal relationship in which epithelial cells in infected mice make IL-23, which can act on IELs to secrete IL-22; IL-22 promotes anti-microbial peptide synthesis and secretion by epithelial cells, particularly Paneth cells. 

Like *Salmonella*, pathogenic strains of *E. coli* cause gastrointestinal disease world-wide and effective treatments and vaccines are highly sought after. When analyzing the pathogenesis of enteroaggregative *Escherichia coli* in enteroids, researchers observed unique patterns of adherence occurring in difference segments of the intestines taken from different donors [44]. This makes it difficult to truly confirm whether certain responses that are due to different cells throughout the enteroid and are derived from multiple donors are driven by a common mechanism of pathogenesis by the microbe or whether it is due to the genetic variability of the enteroids. Additionally, since genetics can play a role in immune responses, should the pathogenesis of these enteric microbes be analyzed focusing on additional factors such as demography or race? The development of treatments for gastrointestinal infections have been limited due to the lack of understanding about the pathogenesis of intestinal organisms; however, if host specificity is a factor that needs to be considered, can information on microbial pathogenesis obtained from human enteroid models be used for directing treatments? It does appear that using organoids will lead to a better understanding of the innate immune signaling triggered in response to a pathogenic infection in the epithelium. 

## 8. Are Organoids the End of Intestinal Epithelial Cell Model Evolution?

As attractive as organoids are in recapitulating the native epithelium and potential applications, there are some noteworthy limitations that need to be considered when interpreting research outcomes. One concern about organoid development is reproducibility; it is not known whether all organoids grown under the same conditions will have the same phenotype [45]. Unlike cell lines of a single genotype, there is also no standard organoid against which new organoid models can be compared. As a result, researchers growing human organoids that show unique phenotypic traits are unable to verify whether their results are reproducible or representative of “human” biology. What by some measures is a strength, the potential for “personalized medicine” drawing on the unique genetics of a donor can become a weakness in appreciating whether a particular response represents “normal” or not. Despite these issues, if intestinal organoid cultures are not the end of this evolution, organoids are certainly the leading edge. New and significant advances in our understanding of the host intestine–microbe relationship based on the use of organoids have already been reported and more are certain to come.

## Figures and Tables

**Figure 1 microorganisms-07-00406-f001:**
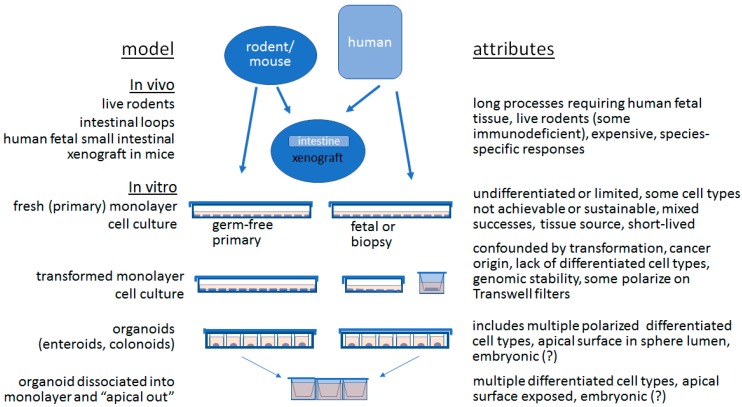
Attributes of popular model systems for studying intestinal epithelial cells, depicted as evolving with improving refinement in ex vivo cell resemblance to native epithelium.

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
