# Peer review of "Opinion: Are Organoids the End of Model Evolution for Studying Host Intestinal Epithelium/Microbe Interactions?"

_microorganisms, 2019, doi:10.3390/microorganisms7100406_

Round 1

Reviewer 1 Report

This is an elegantly written review by George et al on organoids as a model to study host-intestinal epithelium-microbe interactions, with insightful overviews of the limitations and advantages of organoids as a model system to study the biology of intestinal epithelium. Below are my suggestions to improve the manuscript.

The section “ 4. Intestinal organoids closely resemble native epithelium” needs to be expanded with more details. For example, line 32 – what are the ion transport systems referred here”? Line 34 – what are the “different properties” along the length of the intestinal segments?

Suggestion for organizational changes –

-move section #7 to the end of the review as a summary

-change sections #9 (virus), 10 (virus), 11 and 12 (bacteria) as sub-sections within section# 8 “Organoids and Microbes”

      3) Section 8, lines 15-32. Please a brief description on how lacking immune cells in the organoid would limit the study of immune response to microbes, since in vivo there should be crosstalk between immune cells and epithelial cells in the response to microbes.

Section 2 “Cell biology of organoids” line 31 – “secretion, filtration, absorption, endocrine secretion, neural activity and contraction” – are these functions retained in organoids after differentiation of intestinal stem cells? Please clarify and elaborate. Section 3 “Native intestinal epithelial cell organization” line 17 – “other niche factor”. What are these factors? Please specify and give examples. Line 12 – please remove “thought to”.

Reviewer 2 Report

In this paper, the intentions of the authors would be those to describe the role of organoids as an evolutionary model for the study of host intestinal epithelium/microbe interaction. The topic is very interesting considering the growing scientific interest and the numerous papers recently published in these last years. Albeit the theme is quite extensive considering that intestine tissue is a complex structure with several functional aspects, the authors should organize better the manuscript in the chapters and adding other information as well as most updated references. In addition the manuscript doesn’t reveal particular element of novelty considering that other papers have been published on this topic (es. Cell Mol Gastroenterol Hepatol. 2016, 19;3(2):138-149. doi: 10.1016/j.jcmgh.2016.11.007) not cited by the . eCollection 2017 Mar.Gastrointestinal Organoids: Understanding the Molecular Basis of the Host-Microbe Interface.Hill DR1, Spence JR2. Moreover, a graphical abstract should be added to better explain and better describe to the readers the evolutionary and positive aspects of these structures respect to their potentiality in studying intestinal epithelium/microbe interactions. The Figure 1 used by the authors doesen’t seem very clear.

Regarding the chapter 3 and intestinal epithelial cell organization some references regarding the integrity of intestinal barrier and the role of aquaporins should be added (Zhang et al., 2011 doi.org/10.1016/j.febslet.2011.08.045; Pelagalli et al., 2016 doi:10.3390/ijms17081213). Moreover the chapters indicated as 9., 10. 11., 12. should be included as subparagraphs of chapter 8 (Organoids and microbes).

A better description of intestine barrier polarity and its role in studying the interaction with microbe should be added to the actual text.

References as: 1. Co et al., 2019, Cell Reports 26, 2509–2520; 2. Costa J et al., 2019, Front Bioeng Biotechnol.  18;7:144, 3. Fair KL et al., 2018, Philos Trans R Soc Lond B Biol Sci. 373(1750). pii: 20170217; 4. Yin Y and Zhou D, 2018, Front Cell Infect Microbiol. 8:257. doi: 10.3389/fcimb.2018.00257. 2018 should be added.

Round 2

Reviewer 2 Report

now the manuscript is suitable for publication